

# The impact of doctor–patient communication on medication adherence and blood pressure control in patients with hypertension: a systematic review

Jianwei Zeng[1], Yuqiang Gao[2], Chen Hou[3] and Tao Liu[4]

[1] Department of Cardiovascular Medicine, The Central Hospital of Yongzhou, University of South China, Yongzhou, Hunan, China
[2] Department of Clinical Laboratory, Zaozhuang Municipal Hospital, Zaozhuang, Shandong, China
[3] Department of Neurology, The Central Hospital of Yongzhou, University of South China, Yongzhou, Hunan, China
[4] "The 14th Five-Year Plan" Application Characteristic Discipline of Hunan Province (Clinical Medicine), Faculty of Clinical Medicine, Changsha Medical University, Changsha, Hunan, China

Corresponding author
Tao Liu, tao_liu_terry@163.com

## ABSTRACT

**Background:** This systematic review aims to present existing evidence concerning the effects of doctor–patient communication on medication adherence and blood pressure control in hypertensive patients.

**Methods:** Two researchers independently conducted comprehensive searches of five databases and screened relevant studies published from the inception of these databases up to July 21, 2024. The titles, abstracts, and full texts of all the retrieved articles subsequently underwent rigorous duplicate screening, according to predefined inclusion and exclusion criteria. We then synthesized the findings in a narrative format of the included studies. Finally, two researchers independently assessed study quality.

**Results:** Eighteen observational studies encompassing 21,542 patients and seven randomized controlled trials with 2,804 patients were included in the systematic review. Diverse approaches were employed for evaluating doctor–patient communication and medication adherence in these studies, with identified themes including communication content and communication skills. Various facets of doctor–patient communication, including patient satisfaction with doctor–patient communication, the specific content discussed, the style of communication, the comprehensive communication skills of doctors, and the duration of these conversations, were scrutinized. In general, the results suggest a promising link between effective doctor–patient communication and increased medication adherence and blood pressure control. Nonetheless, the presence of nuanced variations and subtle distinctions within the literature underscores the imperative for deeper exploration and consideration in clinical practice. Additionally, effective communication interventions must attain a certain threshold of intensity and endure for an adequate duration.

**Conclusion:** This review underscores the pivotal role of robust doctor–patient communication in improving both medication adherence and blood pressure control. Nevertheless, additional research may be warranted to address the disparities

and subtleties in the literature and to establish precise implications for clinical practice. Moreover, in clinical practice, strategies to enhance doctor–patient communication should be incorporated, given the potential to improve medication adherence and blood pressure control among hypertensive patients.

**Trial registration:** Systematic review registration: PROSPERO with registration number CRD42024503112.

# INTRODUCTION

Hypertension is one of the most common cardiovascular diseases and is responsible for numerous premature deaths worldwide (*Fu et al., 2023*). It is often referred to as a "silent killer" and typically presents with no overt symptoms, making it a covert yet significant health threat. In September 2023, the World Health Organization released the Global Report on Hypertension, which revealed that the global age-standardized prevalence of hypertension remained relatively stable between 1990 and 2019, increasing slightly from 32% to 33%. However, the number of adults with hypertension doubled during this period, rising from 650 million in 1990 to 1.3 billion in 2019 (*World Health Organization, 2023*).

Regrettably, despite this notable increase in cases, the rates of hypertension control and treatment continue to lag behind, remaining at suboptimal levels (*Yin et al., 2022*; *Zhao, 2021*). For example, in China, the awareness, treatment, and control rates of hypertension are 51.5%, 46.1%, and 16.9%, respectively (*Yin et al., 2022*). Globally, 59.0% of women and 49.0% of men with hypertension reported a previous diagnosis of hypertension in 2019, and 47.0% of women and 38.0% of men were under treatment. The control rates among people with hypertension in 2019 were 23.0% for women and 18.0% for men (*NCD Risk Factor Collaboration (NCD-RisC), 2021*). Discrepancies between treatment rates and control rates result from various factors (*Choudhry et al., 2022*; *Wang et al., 2023*), effective doctor–patient communication plays a crucial role in hypertension control.

Efficient communication is pivotal at all stages of healthcare. It is indispensable not only during the process of obtaining medical history for precise diagnosis but also in articulating treatment plans in a way that patients can readily understand and adhere to. Moreover, establishing an open channel of communication is crucial to guarantee that patients feel comfortable reaching out with any questions or concerns even after their appointment (*Voogt, Pratt & Rollet, 2022*). Medication adherence pertains to the degree of alignment between a patient's medication-taking behavior and the physician's prescriptions (*Georges et al., 2022*). The National Health and Nutrition Examination Survey revealed a strong association between medication adherence and hypertension control in American hypertensive patients necessitating long-term medication (*Centers for Disease Control and Prevention (CDC), 2012*). In a meta-analysis of 25 studies involving 12,603 patients, the 8-item Morisky Medication Adherence Scale (MMAS-8) was utilized as a tool to evaluate medication adherence. Scores below 6 on the MMAS-8 were classified

as indicative of nonadherence, suggesting potential challenges in following prescribed medication regimens. A total of 45.2% of hypertensive patients demonstrated poor medication adherence. Notably, patients with good medication adherence presented a lower incidence of uncontrolled blood pressure (Abegaz et al., 2017). Additionally, poor medication adherence was also associated with an increased risk of cardiovascular disease events (Lee et al., 2021). Medication adherence is a multifaceted event influenced by various factors, including the patient, the healthcare team, government policies, and the socioeconomic context (Choudhry et al., 2022; Qin et al., 2020). This phenomenon can be attributed to different stages of treatment, including prescription, regimen adjustments, and maintenance (Choudhry et al., 2014). Effective communication with patients is a crucial skill for doctors throughout these stages (Ge et al., 2022).

Notably, there have been no relevant systematic reviews investigating the connection between doctor–patient communication and medication adherence in patients with hypertension. Additionally, no conclusions have been drawn regarding whether doctor–patient communication affects blood pressure. Nonetheless, its potential harm is substantial, necessitating extended management and strict lifestyle interventions. In this systematic review, we aimed to present existing evidence concerning the effects of doctor–patient communication on medication adherence and blood pressure control in hypertensive patients.

## METHOD

This manuscript was written in line with the Preferred Reporting Items for Systematic Reviews and Meta-Analyses (PRISMA) guidelines (Hutton et al., 2015; Välimäki et al., 2021). We have registered in the PROSPERO International Prospective Register of Systematic Reviews (CRD42024503112). A protocol was not prepared.

### Searching strategies

To mitigate retrieval bias, we deployed multiple search strategies. This process involved comprehensive literature searches conducted by two researchers, JWZ and CH, across five prominent English databases: PubMed, Embase, the Cochrane Library, EBSCO, and Web of Science. Our search spanned from the inception of these databases up to July 21, 2024, without any other restrictions. The titles, abstracts, and full texts were subsequently subjected to rigorous duplicate screening, which was performed by the same researchers, JWZ and YQG, according to predefined inclusion and exclusion criteria. In cases of disagreements, we sought the expertise of a third researcher, TL. The search terms used are shown in the Supplemental Materials.

### Study selection process

After the database search was conducted, all identified literature was first imported into the reference management software EndNote to remove duplicate citations. Two independent researchers (JWZ and YQG) then screened the titles and abstracts of the remaining literature, excluding any that clearly did not meet the inclusion criteria. In cases of disagreement between the two researchers, a third researcher (LT) was consulted to make

the final decision. For studies that passed the initial screening, the full texts were retrieved for further assessment.

The included studies included a range of designs, including observational studies (both cross-sectional and cohort) and randomized controlled trials (RCTs). The inclusion criteria were as follows: (a) aged 18 years or older, (b) study population consisting of individuals with hypertension, and (c) an assessment of the impact of doctor–patient communication on blood pressure management or medication adherence. To be included, studies had to meet all three criteria. Studies were excluded if they lacked sufficient data, had irrelevant outcomes, or were commentaries, conference abstracts, reviews, letters, case reports, non-English literature, or animal studies.

## Data extraction and recording

To minimize bias, two researchers (JWZ and YQG) independently extracted and recorded data from the included studies *via* Microsoft Excel. We designed a standardized data extraction form, which includes the following information: author's name, study location, study design, publication year, study population characteristics, and sample size. In the case of observational studies, additional data included details on the measurement methods for communication and medication adherence. For RCTs, the additional data included information on communication training types and medication adherence measurement methods. If disagreements arose, a third researcher (TL) was consulted, and the TL was responsible for making the final decision. Finally, two researchers (JWZ and YQG) synthesized the findings in narrative format.

## Quality assessment

To evaluate the quality of the included studies, two researchers (JWZ and CH) independently assessed study quality *via* the Newcastle–Ottawa Scale (NOS) for cohort studies (*Stang, 2010*), the Agency for Healthcare Research and Quality (AHRQ) checklist for cross-sectional studies (*Sheleme, Bekele & Ayela, 2020*), and the Cochrane bias tool for RCTs (*Chai et al., 2023*; *Higgins et al., 2024*). Following a pilot phase, a review leader (TL) assessed the risk of bias for each study and raised any questions or concerns about the included studies to the review team for discussion and consensus. The NOS consists of 9 items, with a score ranging from 7–9 points indicating high-quality research, 4–6 points indicating medium-quality research, and less than 4 points indicating low-quality research. The AHRQ list comprises 11 items. Quality scores of 0–3, 4–7, and 8–11 indicate that the cross-sectional studies are rated as having high, medium, and low risk of bias, respectively. In the case of the Cochrane bias tool, if at least 4 of the 7 items in total for RCTs are assessed as having low-risk bias, then the overall risk of bias is considered low.

## Data synthesis

To address the heterogeneity among the studies, we adopted a narrative approach to synthesize the data, presenting information in text and tables that corresponded to each of the review objectives.
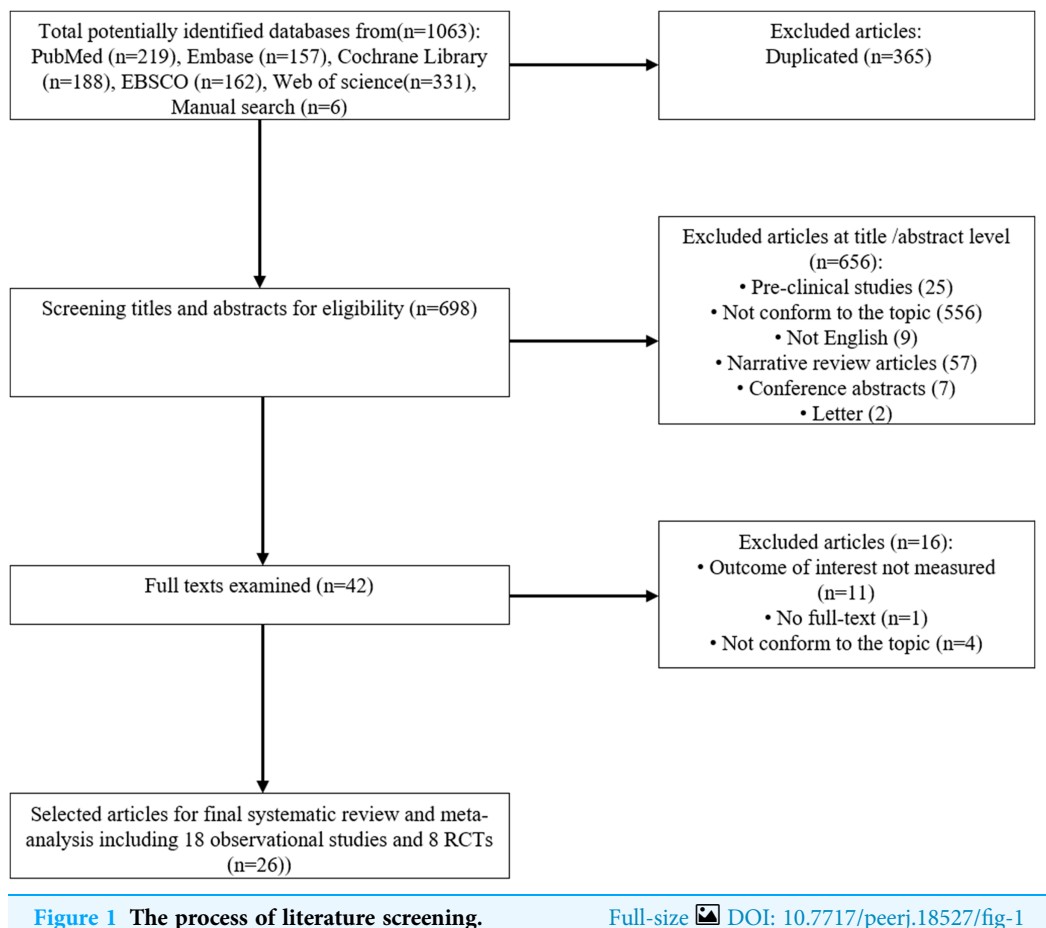

**Figure 1** The process of literature screening.

## RESULTS

### Selection of studies

The process of study selection is illustrated in Fig. 1. We initially retrieved 1,063 records by searching the five electronic databases and conducting manual searches. After removing duplications, a total of 697 records remained for the initial screening. Two researchers screened the titles and abstracts, resulting in 41 studies selected for full-text review. Sixteen of these studies were excluded for various reasons, such as the outcome of interest not being measured, lack of full-text, and not conforming to the topic. Ultimately, we included eighteen observational studies involving 21,542 patients and seven randomized controlled trials with 2,804 patients in the systematic review.

### Characteristics of the included studies

This review included eighteen observational studies (*Amin et al., 2018*; *Chang et al., 2021*; *Cho et al., 2007*; *Harris et al., 1995*; *Kressin et al., 2007*; *Mahmoudian et al., 2017*; *Martin et al., 2010*; *Naik et al., 2008*; *Schoenthaler et al., 2009*, *2017*, *2018*, *2016*; *Świątoniowska-Lonc et al., 2020*; *Tamblyn et al., 2010*; *Turner et al., 2009*; *Woojung et al., 2017*; *Yoshida et al., 1995*; *Zanatta et al., 2020*) and seven RCTs (*Cooper et al., 2011*; *Escortell-Mayor et al., 2020*; *Kressin et al., 2016*; *Manze et al., 2015*; *Qureshi et al., 2007*; *Tavakoly Sany et al., 2020*;

**Table 1  Characteristics of the included observational studies.**

| Study | Area | Design | Patients | Age (Mean/Mean ± SD/%) | Gender | Communication measure | Medication adherence measure | Time to assess |
|---|---|---|---|---|---|---|---|---|
| Yoshida et al. (1995) | Japan | Cross-sectional | 365 | 70 ± 5.9 | Women 53.5% | Rating by patients; assessment of patient-hospital relationship | The quality of life scale for elderly Japanese patients with cardiovascular diseases | / |
| Harris et al. (1995) | USA | Cross-sectional | 333 | 69 ± 11 | Women 77.0% | Rating by patients; patient satisfaction questionnaire | A locally developed and tested medication assessment instrument | / |
| Kressin et al. (2007) | USA | Cross-sectional | 793 | White: 64.4 African-American: 67.3 | / | Rating by patients; assessment of doctor–patient interaction | Items from 2 well-validated measures from other literature | / |
| Cho et al. (2007) | USA | Cross-sectional | 554 | 63.1 ± 11.2 | Women 2.0% | Rating by patients; a validated 3-item measure developed by Kaplan and Greenfield | 4-item self-report instrument | / |
| Naik et al. (2008) | USA | Cross-sectional | 212 | Controlled BP group: 66.4 ± 8.4  Uncontrolled BP group: 67.4 ± 9.2 | Controlled BP group: women 2.0%  Uncontrolled BP group: women 1.0% | Rating by patients; patient–clinician communication assessment items adapted from subscales of the patient assessment of chronic illness care scale | A self-report item; a longitudinal medication refill adherence measure | / |
| Schoenthaler et al. (2009) | USA | Cross-sectional | 439 | 57.69 ± 12.1 | Women 68.0% | Rating by patients; 13-item follow-up communication scale | Morisky medication adherence scale | / |
| Turner et al. (2009) | USA | Cross-sectional | 202 | 70–79:67.2% >80:32.8% | Women 65.9% | Rating by patients; assessment of doctor–patient interaction | Assessed by a question adapted form Morisky scale | / |
| Martin et al. (2010) | USA | Cross-sectional | 434 | / | Women 67.5% | Rating by patients; provider-patient relationship | Self-reported medication-taking behavior | / |
| Tamblyn et al. (2010) | Canada | Cohort | 13,205 | 61.4 ± 13.5 | Women 57.7% | Rating by examinations; medical council of Canada licensing examination scores | Means of records of prescription claims from all community-based pharmacies | 6 months |
| Schoenthaler et al. (2016) | USA | Cohort | 815 | 57.0 ± 12.2 | Women 70.9% | Rating by patients; patients' perception of the quality of their providers' communication and the extent to which the provider encourages patient participation in the treatment process | Morisky medication adherence scale | 12 months |
| Schoenthaler et al. (2017) | USA | Cohort | 92 | 59.7 ± 10.6 | Women 57.6% | Rating by the medical interaction process system | Electronic monitoring device. | 3 months |
| Mahmoudian et al. (2017) | Iran | Cross-sectional | 300 | <65:66.3% ≥65:33.7% | Women 71.0% | Rating by patients; patient's satisfaction questionnaire | Morisky medication adherence scale | / |
| Woojung et al. (2017) | USA | Cross-sectional | 191 | <64:18.0% 65–75:48.7% >76:33.0% | Women 77.5% | Rating by patients; communication and interpersonal treatment subscales of the primary care assessment survey | Morisky medication adherence scale | / |

| Study | Area | Design | Patients | Age (Mean/Mean ± SD/%) | Gender | Communication measure | Medication adherence measure | Time to assess |
|---|---|---|---|---|---|---|---|---|
| Amin et al. (2018) | Dhaka | Cross-sectional | 253 | 49.15 ± 10.375 | Women 45.1% | Rating by patients; the communication assessment tool | Morisky Medication Adherence Scale | / |
| Schoenthaler et al. (2018) | USA | Cohort | 75 | 59.93 ± 10.29 | Women 56.0% | Rating by patients; the control preferences scale | Electronic monitoring device | 3 months |
| Świątoniowska-Lonc et al. (2020) | Poland | Cross-sectional | 250 | 61.23 ± 14.34 | Women 44.0% | Rating by patients; the communication assessment tool | The adherence to refills and medication scale | / |
| Zanatta et al. (2020) | Italy, Poland | Cross-sectional | 458 | 72.4 ± 7.8 | Women 60.7% | Rating by patients; the communication assessment tool | MGLS; ARMS; INAS; ASonA* | / |
| Chang et al. (2021) | USA | Retrospective cohort study | 2,571 | Low communication: 18–44:17.0% 45–64:52.0% ≥65:31.0% High communication: 18–44:15.0% 45–64:52.0% ≥65:33.0% | Low communication: Women 64.0% High communication: Women 63.0% | Rating by patients; adapted from the health plan version of consumer assessment of healthcare providers and systems | Medication refill adherence | 2 years |

**Note:**

*Abbreviations: MGLS, Morisky Green Levine Scale; ARMS, Adherence to Refills and Medications Scale; INAS, Intentional Nonadherence Scale; ASonA, Antecedents and Self-Efficacy on Adherence Schedule (ASonA-SE, self-efficacy subscale).

**Table 2 Characteristics of the included RCTs.**

| Study | Area | Doctors | Patients | Age (Mean/Mean ± SD/%) | Gender | Intervention | Adherence measure |
|---|---|---|---|---|---|---|---|
| Qureshi et al. (2007) | USA | General practitioners (78) | 178 | 55.3 ± 10.14 | Women 62.5% | Training in management of hypertension | Data from the electronic medication event monitoring system |
| Cooper et al. (2011) | USA | Primary care physicians (41) | 138 | *E: 59.7 ± 11.9<br><br>C: 62.4 ± 12.1 | *E: Women 65.1%<br><br>C: Women 61.8% | Physician communication skills training and patient coaching by community health workers. | Morisky medication adherence scale |
| Tinsel et al. (2013) | German | General practitioners (37) | 1,120 | *E: 63.8 ± 12.1<br><br>C: 65.0 ± 12.4 | *E: Women 53.3%<br><br>C: Women 55.3% | Training in shared decision making | Medication adherence report scale |
| Manze et al. (2015) | USA | Providers in primary care clinics (58) | 203 | 21–30 0.5%<br>31–40 2.0%<br>41–50 16.3%<br>51–60 32.0%<br>61–70 29.6%<br>71–80 15.3%<br>81–90 4.4% | Women 72.4% | Two separate workshops related to patient-centered counseling and cultural competency | Hill-Bone compliance to high blood pressure therapy scale, |
| Kressin et al. (2016) | USA | Primary care providers (29) | 514 | 66.2 | Women 1.2% | Training in communication skills | Items from 2 well-validated measures from other literature |
| Tavakoly Sany et al. (2020) | Iran | Physicians (35) | 240 | 54.8 ± 11.5 | Women 77.34% | Training in communication skills | Adult primary care questionnaire |
| Escortell-Mayor et al. (2020) | Spain | Family doctors and nurses | 411 | 55.3 ± 6.7 | Women 51.6% | Education and coronary risk evaluation | Morisky-Green questionnaire |

**Note:**
* Abbreviations: E, experimental group; C, control group.

Tinsel et al., 2013). The studies were conducted between 1995 and 2020, primarily in the United States. The study population included two categories of professionals: physicians and primary care providers, such as physician assistants, nurse practitioners, and clinical pharmacists, in general practices. The majority of the study participants were middle-aged or older adults. The proportion of female participants in the studies ranged from 1.2% to 77.5%. In the studies by Naik et al. (2008), Cho et al. (2007), and Kressin et al. (2016), the study populations consisted of military veterans. Notably, the majority of military veterans are male, with a very low proportion of females. Cross-sectional studies ($n = 13$) examined data at specific points in time, whereas five cohort studies had follow-up periods ranging from 3 months to 2 years. Additionally, the included RCTs reported outcomes at intervals between 6 weeks and 18 months postintervention. The details are displayed in Tables 1 and 2.

## Quality assessment

We identified that all RCTs were at high risk of bias in terms of blinding due to the nature of the intervention, which made blinding in communication training for doctors

challenging. Additionally, only two studies reported on the method of allocation concealment (Fig. 2). Among the cross-sectional studies, six were classified as having a low risk of bias, and seven were characterized as having a moderate risk of bias (as detailed in Table 3). As assessed *via* the NOS, the overall NOS scores for each included cohort study ranged from seven to nine stars (as shown in Table 4).

## Evaluation and intervention in doctor–patient communication

Among the observational studies, two studies (*Schoenthaler et al., 2017*; *Tamblyn et al., 2010*) employed an objective measure (standardized test scores or medical interaction process system) to evaluate doctor–patient communication, whereas the remaining studies relied on self-reported measures such as scales and questionnaires, which relied on patient ratings (predominantly) or tests (to a lesser extent), including the assessment of doctor–patient interaction, Patient Satisfaction Questionnaire, and Medical Council of Canada licensing examination scores. Furthermore, two studies examined collaborative communication between healthcare providers and patients (*Naik et al., 2008*; *Schoenthaler et al., 2016*), whereas five studies focused on assessing decision-making styles in doctor–patient interactions (*Chang et al., 2021*; *Cho et al., 2007*; *Mahmoudian et al., 2017*; *Naik et al., 2008*; *Schoenthaler et al., 2018*). Additionally, one study was dedicated to examining the impact of communication etiquette on hypertension management (*Harris et al., 1995*). Various aspects of doctor–patient communication have been examined, including patient satisfaction with doctor–patient communication (*Amin et al., 2018*; *Chang et al., 2021*; *Harris et al., 1995*; *Mahmoudian et al., 2017*; *Martin et al., 2010*; *Świątoniowska-Lonc et al., 2020*; *Yoshida et al., 1995*; *Zanatta et al., 2020*), the specific content discussed (particularly regarding hypertension and medication adherence) (*Kressin et al., 2007*; *Schoenthaler et al., 2017*; *Woojung et al., 2017*), the style of communication (*Chang et al., 2021*; *Cho et al., 2007*; *Mahmoudian et al., 2017*; *Naik et al., 2008*; *Schoenthaler et al., 2009*, *2017*, *2018*, *2016*; *Turner et al., 2009*), the overall communication skills of doctors (*Tamblyn et al., 2010*), and the duration of conversations (*Amin et al., 2018*).

Among the RCTs, *Cooper et al. (2011)* simultaneously intervened with both patients and doctors to enhance communication. Two studies employed an evidence-based patient-centered counseling approach to provide communication skills training for healthcare professionals (*Kressin et al., 2016*; *Manze et al., 2015*), whereas a single study introduced shared decision-making training exclusively to doctors in the intervention group (*Tinsel et al., 2013*). *Escortell-Mayor et al. (2020)* utilized this method. This study focused on a primary healthcare information intervention designed to communicate cardiovascular risk to patients with poorly controlled hypertension. The intervention was implemented at primary healthcare centers, where the intervention group was initially presented with the "low-risk Systematic Coronary Risk Evaluation (SCORE) table" along with striking visual aids and informative brochures aimed at promoting and maintaining optimal cardiovascular health. In the study by *Tavakoly Sany et al. (2020)*, eligible physicians in the intervention group were invited to undergo training aimed at enhancing their communication skills with hypertensive patients. This intervention consisted of three sessions of focus group discussions and two workshops. In the study

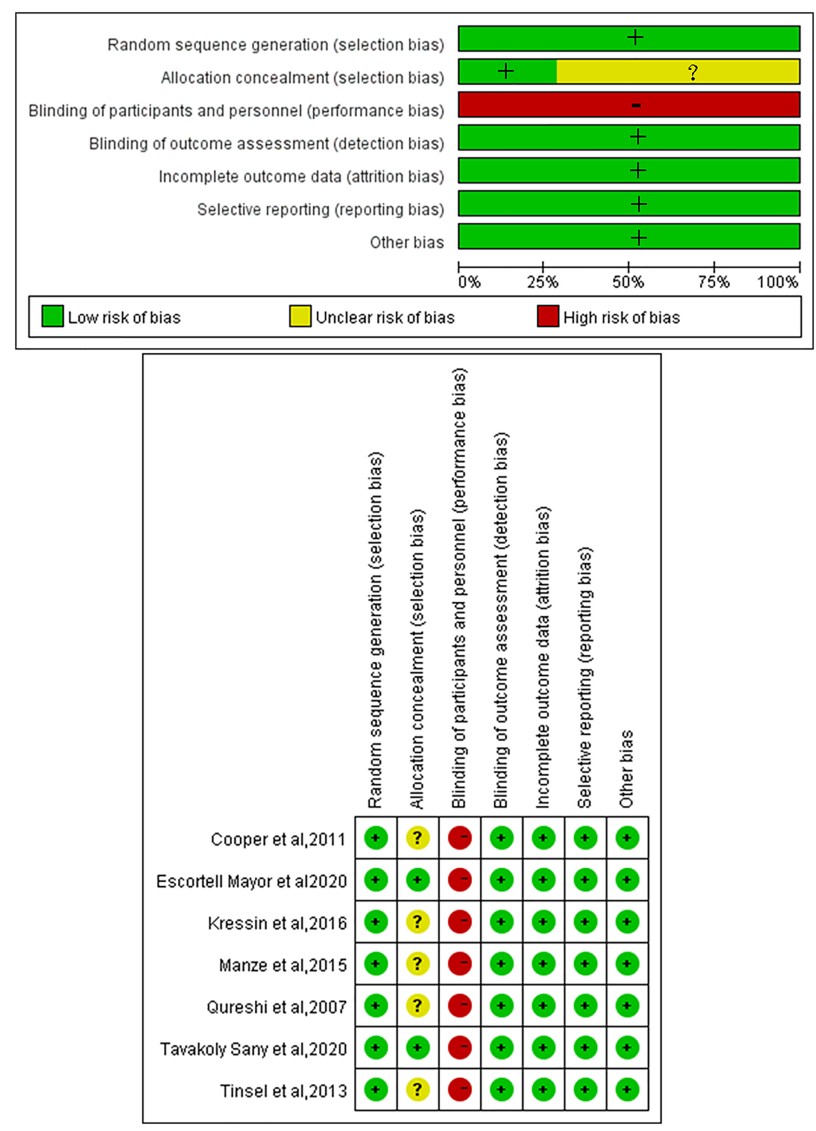

**Figure 2 Quality assessment of RCTs.** +, low risk; −, high risk; ?, unclear risk (*Cooper et al., 2011*; *Escortell-Mayor et al., 2020*; *Kressin et al., 2016*; *Manze et al., 2015*; *Qureshi et al., 2007*; *Tavakoly Sany et al., 2020*; *Tinsel et al., 2013*).                               

by *Qureshi et al. (2007)*, all general practitioners were invited to attend a one-day intensive training session on hypertension, with a specific focus on the utilization of standard treatment algorithms for hypertension management.

## Evaluation in medication adherence

The medication adherence measurements used in the included studies included self-reported tools such as the Morisky Medication Adherence Scale, the Adherence to Refills and Medications Scale, and self-developed questionnaires, among others. The subjective evaluation tools have the advantage of being simple and easy to administer, making them useful tools for quickly assessing adherence in clinical practice. However,

**Table 3 Quality assessment results of cross-sectional studies.**

| Studies | The agency for healthcare research and quality (AHRQ) methodology checklist | | | | | | | | | | | |
|---|---|---|---|---|---|---|---|---|---|---|---|---|
| | Item 1 | Item 2 | Item 3 | Item 4 | Item 5 | Item 6 | Item 7 | Item 8 | Item 9 | Item 10 | Item 11 | Scores |
| *Yoshida et al. (1995)* | Yes | No | No | Yes | Unclear | Yes | Unclear | Yes | No | Yes | No | 5 |
| *Harris et al. (1995)* | Yes | Yes | No | Yes | Unclear | Unclear | Unclear | Yes | Yes | No | No | 5 |
| *Kressin et al. (2007)* | Yes | Yes | Yes | Yes | Unclear | Yes | Yes | Yes | No | Yes | No | 8 |
| *Cho et al. (2007)* | Yes | Yes | Yes | Yes | Yes | Yes | Yes | Yes | No | Yes | No | 9 |
| *Naik et al. (2008)* | Yes | Yes | Yes | Yes | Yes | Yes | Yes | Yes | Yes | Yes | No | 10 |
| *Schoenthaler et al. (2009)* | Yes | Yes | Yes | Yes | Unclear | Yes | Yes | Yes | Yes | Yes | No | 9 |
| *Turner et al. (2009)* | Yes | Yes | Yes | Yes | Yes | Yes | Yes | Yes | Yes | Yes | No | 10 |
| *Martin et al. (2010)* | Yes | Yes | Yes | Yes | Unclear | Yes | No | No | No | Yes | No | 6 |
| *Mahmoudian et al. (2017)* | Yes | Yes | Yes | Yes | Unclear | Unclear | No | Yes | No | Yes | No | 6 |
| *Woojung et al. (2017)* | Yes | No | Yes | Yes | Yes | Yes | No | Yes | No | Yes | No | 7 |
| *Amin et al. (2018)* | Yes | No | Yes | Yes | Unclear | Yes | No | Yes | Yes | No | No | 6 |
| *Świątoniowska-Lonc et al. (2020)* | Yes | Yes | Yes | Yes | Yes | Yes | No | No | Yes | Yes | No | 8 |
| *Zanatta et al. (2020)* | Yes | Yes | Yes | Yes | Unclear | Yes | Yes | No | No | Yes | No | 7 |

Notes:
Item 1: Define the source of information (survey, record review);
Item 2: List inclusion and exclusion criteria for exposed and unexposed subjects (cases and controls) or refer to previous publications;
Item 3: Indicate the time period used for identifying patients;
Item 4: Indicate whether or not the subjects were consecutive if not population-based;
Item 5: Indicate whether evaluators of the subjective components of the study were masked to other aspects of the status of the participants;
Item 6: Describe any assessments undertaken for quality assurance purposes (*e.g.*, test/retest of primary outcome measurements);
Item 7: Explain any patient exclusions from analysis;
Item 8: Describe how confounding was assessed and/or controlled;
Item 9: If applicable, explain how missing data were handled in the analysis;
Item 10: Summarize patient response rates and completeness of data collection;
Item 11: Clarify what follow-up, if any, was expected and the percentage of patients for which incomplete data or follow-up data were obtained.

**Table 4 Quality assessment results of the cohort studies.**

| Study | Newcastle–Ottawa (NOS) scale | | | | | | | | |
|---|---|---|---|---|---|---|---|---|---|
| | Item 1 | Item 2 | Item 3 | Item 4 | Item 5 | Item 6 | Item 7 | Item 8 | Total |
| *Tamblyn et al. (2010)* | 1 | 1 | 1 | 1 | 2 | 1 | 1 | 0 | 8 |
| *Schoenthaler et al. (2016)* | 1 | 1 | 1 | 1 | 2 | 1 | 1 | 1 | 9 |
| *Schoenthaler et al. (2017)* | 1 | 1 | 1 | 1 | 2 | 1 | 1 | 0 | 8 |
| *Schoenthaler et al. (2018)* | 0 | 1 | 1 | 1 | 2 | 1 | 1 | 0 | 7 |
| *Chang et al. (2021)* | 1 | 1 | 1 | 0 | 2 | 1 | 1 | 1 | 8 |

Notes:
Item 1: Representativeness of the exposed cohort (1 point).
Item 2: Selection of the nonexposed cohort (1 point).
Item 3: Ascertainment of exposure (1 point).
Item 4: Demonstration that the outcome of interest was not present at the start of the study (1 point).
Item 5: Comparability (2 points).
Item 6: Assessment of outcome (1 point).
Item 7: Follow-up long enough for outcomes to occur (1 point).
Item 8: Adequacy of follow-up of cohorts (1 point).

their limitations lie in the potential influence of patients' social desirability bias, where patients may report answers that they believe the researchers want to see, rather than their actual adherence behaviors.

Nevertheless, a limited number of studies have employed objective measures, such as prescription records and data from the electronic medication event monitoring system, to assess patient medication adherence (*Schoenthaler et al., 2017*, *2018*). These objective measurement methods provide direct and verifiable data, which helps to enhance the accuracy of adherence assessments. For instance, electronic pill boxes can record the date and time of each opening, monitoring the distribution and use of medication. However, the cost of objective measurement methods may be high, and there may be challenges in implementation. For example, the cost of electronic medication event monitoring systems limits their application in large-scale.

## Relationships between doctor–patient communication and blood pressure

The connection between doctor–patient communication and blood pressure was assessed in two cross-sectional studies (*Cho et al., 2007*; *Naik et al., 2008*) and six RCTs (*Cooper et al., 2011*; *Escortell-Mayor et al., 2020*; *Kressin et al., 2016*; *Manze et al., 2015*; *Tavakoly Sany et al., 2020*; *Tinsel et al., 2013*). *Naik et al. (2008)* reported a significant relationship between decision-making style, proactive communication, and collaborative communication with hypertension control. Using a structural equation model with path analysis, this study explored the total, direct, and indirect effects of each variable on hypertension control outcomes. The findings revealed that decision-making style ($\beta = 0.20$, $P < 0.01$) and proactive communication ($\beta = 0.50$, $P < 0.001$) had significant direct impacts on hypertension control. Additionally, collaborative communication indirectly influenced hypertension control by directly affecting both decision-making style ($\beta = 0.28$, $P < 0.001$) and proactive communication, highlighting its indirect yet crucial role in effective hypertension management. However, a cross-sectional analysis by *Cho et al. (2007)* was conducted among 554 hypertensive veterans, the adjusted analysis revealed that a lower participatory decision-making style score did not significantly influence blood pressure control (odds ratio, 1.09; 95% confidence interval, 0.99–1.20).

When the included RCTs were analyzed, *Escortell-Mayor et al. (2020)* and *Cooper et al. (2011)* emphasized the favorable effects of communication training on blood pressure regulation. In the study by *Escortell-Mayor et al. (2020)*, an informative intervention was found to enhance blood pressure control in hypertensive patients. The intervention included using the "SCORE table" for low-risk assessment, complemented by impactful imagery highlighting cardiovascular risk (CVR). Informative pamphlets were also distributed, detailing each patient's SCORE table score and offering practical recommendations for promoting optimal cardiovascular health. These findings highlight the effectiveness of the use of visual cues related to CVRs during doctor–patient interactions, which significantly aids patients in achieving effective hypertension management. *Cooper et al. (2011)* demonstrated that training aimed at improving doctor–patient communication skills and patient coaching by community health workers resulted in increased patient-centered engagement, as assessed by patients' perceived involvement in care. Additionally, *Tavakoly Sany et al. (2020)* provided further support for the positive effects of physician communication training, as it positively impacts hypertension

outcomes. Furthermore, *Kressin et al. (2016)* demonstrated that enhanced clinician counseling resulted in an overall improvement in blood pressure, with a particular benefit observed among white patients. Conversely, *Manze et al. (2015)* reported that communication training for primary care physicians did not significantly alter blood pressure control. This suggests that although communication training holds value, it may not consistently result in quantifiable enhancements in patient outcomes. *Tinsel et al. (2013)* concentrated on training general practitioners in shared decision-making (SDM), and the study did not reveal significant effects of SDM training on blood pressure control, suggesting that the influence of SDM training on hypertension outcomes may necessitate additional investigation.

## Relationship between doctor–patient communication and medication adherence

The impact of doctor–patient communication on medication adherence was investigated in 17 observational studies (*Amin et al., 2018*; *Chang et al., 2021*; *Cho et al., 2007*; *Harris et al., 1995*; *Kressin et al., 2007*; *Mahmoudian et al., 2017*; *Martin et al., 2010*; *Schoenthaler et al., 2009*, *2017*, *2018*, *2016*; *Świątoniowska-Lonc et al., 2020*; *Tamblyn et al., 2010*; *Turner et al., 2009*; *Woojung et al., 2017*; *Yoshida et al., 1995*; *Zanatta et al., 2020*) and seven RCTs (*Cooper et al., 2011*; *Escortell-Mayor et al., 2020*; *Kressin et al., 2016*; *Manze et al., 2015*; *Qureshi et al., 2007*; *Tavakoly Sany et al., 2020*; *Tinsel et al., 2013*).

Patient satisfaction with doctor–patient communication is a crucial factor influencing medication adherence. A study by *Zanatta et al. (2020)* found that hypertensive patients who were more satisfied with their doctor's communication demonstrated significantly greater self-efficacy, which, in turn, improved their medication adherence. Similarly, research by *Mahmoudian et al. (2017)* highlights that patient satisfaction with doctor–patient relationships is correlated with medication adherence among hypertensive patients. In a study conducted in Bangladesh, *Amin et al. (2018)* used a communication assessment tool to assess patients' perceived effectiveness of doctor–patient communication and reported a significant correlation between effective communication and medication adherence among hypertensive patients. *Świątoniowska-Lonc et al. (2020)* evaluated the relationship between satisfaction with doctor–patient communication and self-care and adherence among hypertensive patients. The results revealed that greater communication satisfaction was associated with better self-care and medication adherence. *Martin et al. (2010)* also reported that individuals who reported feeling uncomfortable when communicating with healthcare providers were more likely to be nonadherent. *Yoshida et al. (1995)* developed a comprehensive scale to assess patients' relationships with hospitals and reported that the better the doctor-patient relationship is, the greater the patients' quality of life and medication adherence.

Communication style also has an effect on medication adherence among hypertensive patients. *Schoenthaler et al. (2009*, *2016)* highlighted the importance of collaborative doctor–patient communication styles for medication adherence in hypertensive patients. The 2009 cross-sectional study (*Schoenthaler et al., 2009*) revealed that patients' perceptions of collaborative doctor–patient communication were associated with better

medication adherence, particularly among younger patients and those with depressive symptoms. The 2016 study (*Schoenthaler et al., 2016*) included a 12-month follow-up. These findings indicate that social support and collaborative doctor–patient communication are associated with fewer depressive symptoms, suggesting that effective collaborative communication can indirectly enhance medication adherence. Both of these studies emphasized that collaborative and supportive communication by doctors is a crucial factor in improving medication adherence. *Cho et al. (2007)*, *Schoenthaler et al. (2018)*, *Chang et al. (2021)*, and *Mahmoudian et al. (2017)* all examined the relationship between decision-making style and medication adherence in hypertensive patients, but their findings varied. *Cho et al. (2007)* reported no significant association between doctors' participatory decision-making styles and medication adherence. Similarly, *Mahmoudian et al. (2017)* found no significant association between shared decision-making (SDM) and medication adherence. In contrast, *Schoenthaler et al. (2018)* reported a significant association between SDM and medication adherence. *Chang et al. (2021)* reported that high levels of engagement in SDM were significantly associated with better medication adherence. These differing results may be attributed to variations in study design, sample characteristics, measurement tools, and cultural contexts, indicating that the role of decision-making style in improving medication adherence requires further investigation across different patient populations and healthcare settings to assess its consistency and effectiveness. In addition, *Schoenthaler et al. (2017)* found that patient-centered communication styles were significantly associated with medication adherence.

Some studies have underscored the importance of providing comprehensive information in medical communication. *Schoenthaler et al. (2017)* reported that insufficient discussion about patients' sociodemographic information and medication plans was associated with poorer medication adherence. *Kressin et al. (2007)* highlighted that communication regarding specific medication guidelines significantly impacts medication adherence. *Woojung et al. (2017)* revealed that informative communication, such as providing detailed explanations about the reasons for and methods of medication use, could significantly enhance patients' medication adherence and improve their overall medication experience.

The duration of communication and the communication skills of doctors have been proven to be correlated with medication adherence among hypertensive patients. *Amin et al. (2018)* reported that longer communication time between doctors and patients was associated with greater adherence to antihypertensive medications. *Tamblyn et al. (2010)* used scores from the Medical Council of Canada licensing examination to assess doctors' communication skills and reported a significant association between doctors' communication ability and medication adherence.

In the included RCTs, *Escortell-Mayor et al. (2020)* and *Qureshi et al. (2007)* underscored the beneficial impact of tailored communication training. The former demonstrated how an informative intervention improved medication adherence in hypertensive patients, emphasizing the value of tailored communication. The latter reported that specialized training for general practitioners enhanced adherence to antihypertensive drugs. However, *Manze et al. (2015)* reported that communication

training for primary care doctors did not significantly alter medication adherence. *Cooper et al. (2011)* similarly reported that training directed at physician–patient communication skills did not lead to increased medication adherence. *Tinsel et al. (2013)* revealed no significant effects on medication adherence in hypertensive patients, implying a need for further investigation of the impact of SDM training. However, *Tavakoly Sany et al. (2020)* provided additional support for the positive influence of physician communication training by demonstrating improvements in medication adherence and self-efficacy. Additionally, *Kressin et al. (2016)* emphasized that enhanced clinician counseling positively impacts medication adherence, highlighting the potential of such interventions in enhancing patient care.

## DISCUSSION

### Main findings and heterogeneity of the studies

This review included eighteen observational studies and seven RCTs. Various methods have been used to evaluate doctor–patient communication, including an assessment of communication content and communication skills. In general, the results suggest a promising link between effective doctor–patient communication and increased medication adherence. However, certain variations and subtleties in the literature necessitate further investigation and consideration in clinical practice. The inclusion of diverse studies may have contributed to the uncertainty regarding the impact of improved doctor–patient communication in this systematic review. First, the variety of communication skills training courses used in these RCTs encompassed a wide range of educational theories, methods, and assessment modes. Second, researchers have employed a variety of methods to assess both medication adherence and communication. Furthermore, in this review, the follow-up period of each study varied. The heterogeneity in communication skills training courses presents a challenge in drawing definitive conclusions about the overall effectiveness of communication skills training. This variation can result in inconsistent outcomes, making it difficult to determine which specific aspects of the training are most beneficial. The diverse methods used to assess medication adherence and communication further complicate the synthesis of findings across studies. Different assessment tools can also produce varying results, making direct comparisons of outcomes challenging. Thus, to increase the validity of future research, it is crucial to standardize both communication training programs and assessment methods for medication adherence and communication. Establishing clear and consistent protocols will facilitate more accurate comparisons across studies and provide a more robust evidence base.

### Effect of doctor–patient communication on blood pressure control

In general, the majority of the included studies highlighted the positive impact of doctor–patient communication on blood pressure control. Regarding information exchange, effective doctor–patient communication empowers doctors to educate patients, equipping them with a deeper comprehension of various aspects pertaining to hypertension. This includes instructions on how to measure blood pressure, the significance of blood pressure control, and the implications of hypertension for health. All of these aspects contribute to

better blood pressure management by patients. Furthermore, robust doctor–patient communication helps establish trust between the medical practitioner and the patient, subsequently increasing the probability of hypertensive patients accepting and adhering to medical recommendations. Throughout this process, doctors' ability to offer encouragement and display empathy are pivotal factors in cultivating trust among hypertensive patients (*Zhang et al., 2019*).

Some studies did not find significant associations between doctor-patient communication and blood pressure control. The sample in *Cho et al.*'s *(2007)* study consisted of United States veterans, nearly all of whom were male, which may limit the general applicability of the study's findings. Furthermore, many veterans gave the highest possible scores to their primary care providers on the PDM style, potentially leading to a ceiling effect that diminishes the relationship between PDM style and outcomes (*Cho et al., 2007*). In the study by *Manze et al. (2015)*, two educational workshops were conducted for doctors in clinics. The aim of these workshops was to enhance the doctors' communication skills regarding hypertension medication adherence, but these skills may have been oversimplified and lacked sufficient practical changes, thus making it difficult to have a significant impact on blood pressure control. In *Tinsel et al. (2013)* study, the general practitioners allocated to the intervention group participated in a 6-h SDM training program. However, it is conceivable that such a measure might not be entirely adequate.

### Effect of doctor–patient communication and medication adherence

Overall, the findings indicate a promising connection between effective doctor–patient communication and improved medication adherence. Some studies did not find significant associations between doctor-patient communication and medication adherence. In *Manze et al. (2015)* study, two educational workshops were conducted with doctors in clinics. These workshops aimed to improve doctors' communication skills related to hypertension medication adherence, which may be too brief and lacking sufficient practice-level changes to impact medication adherence. In *Cooper et al. (2011)*, the intervention exposure for physicians was limited to one-time administration, and for patients, it was limited to a single in-person contact. Additionally, baseline adherence rates were relatively high, ranging from 58.2% to 68.5% across the four patient groups. Moreover, the high level of loss to follow-up among randomized physicians further limits patient recruitment. In *Tinsel et al. (2013)*, the general practitioners allocated to the intervention group participated in a 6-h SDM training program. A similar issue of high baseline adherence was observed, with mean adherence scores being very high at the beginning of the study for both the intervention (93.9 ± 9.8) and control (93.5 ± 10.1) groups (0 = lowest level, 100 = highest level). To improve patient outcomes, future interventions may require more extensive provider training and broader systematic changes. Additionally, high baseline medication adherence may have influenced the results. (*Cooper et al., 2011*; *Manze et al., 2015*; *Tinsel et al., 2013*).

In fact, doctor–patient communication can be established directly by encouraging patients to openly discuss their concerns (*Mago et al., 2018*), serving as the foundation of medical service delivery (*Du et al., 2020*). Whether at the beginning of an appointment or

during follow-up, the effective communication of doctors is crucial. It is only through such effective communication that positive rapport can be established between hypertensive patients and doctors, enabling both parties to fully and effectively convey their comprehension and preferences. In addition, effective doctor–patient communication plays a crucial role in enhancing patient satisfaction and treatment outcomes. Doctors could build trust and alleviate patient anxiety by actively listening to their concerns, addressing fears, and explaining treatment options and expectations clearly (*Epstein et al., 2005*). Therefore, there is a promising connection between effective doctor–patient communication and improved medication adherence. Future studies should ensure sufficient time and intensity for communication interventions and recognize that such interventions may be more beneficial for patients with lower baseline medication adherence.

We would like to underscore the importance of the shared decision-making style. The adoption of a shared decision-making style is on the rise (*Tan et al., 2024*; *Waddell et al., 2021*). In this model, patients are provided with information about their medical conditions so that they can participate in their own medical decisions (*Morrison et al., 2022*). In this systematic review, *Schoenthaler et al. (2018)* and *Chang et al. (2021)* reported that shared decision-making styles were more likely to lead to improved medication adherence among black individuals than were the findings of *Tinsel et al. (2013)*, *Cho et al. (2007)*, and *Mahmoudian et al. (2017)*, whose studies did not identify such an association. The study population in *Tinsel et al.*'s *(2013)* study consisted of Germans, *Cho et al.*'s *(2007)* study population consisted of United States veterans, and *Mahmoudian et al.*'s *(2017)* study population consisted of Iranians. There could be two reasons for this. First, it is common for black individuals to experience more severe hypertension and/or complications, suggesting that they might particularly benefit from effective communication to enhance medication adherence. Second, healthcare providers can make more effective interventions by actively listening to and respecting the social and cultural perspectives of black patients.

**Implications for practice**

A cross-sectional survey in which a self-report questionnaire was developed and sent to hypertension doctors in Hunan Province, China, between May 1, 2022, and July 1, 2022, also indicated that "heavy clinical work" (84.3%) and "poor doctor–patient communication" (71.2%) were the two main obstacles to improving medication adherence in hypertensive patients (*Liu et al., 2023*). These findings suggest that doctor–patient communication may indirectly impact blood pressure control through the mediating effect of medication adherence. Both doctor–patient communication and medication adherence are complex phenomena influenced by a multitude of factors in clinical practice (*Chandra & Mohammadnezhad, 2021*; *Emadi et al., 2022*), and the impact of communication skills training in RCTs is similarly affected by these aforementioned factors. On the basis of this review, it is recommended that targeted measures be implemented to improve doctor–patient communication among hypertensive patients. These measures should include the following: enhancing doctor–patient communication skills training for doctors and

medical students (*Ding et al., 2020*); providing training programs that focus on improving communication skills; and addressing patients' concerns, explaining the significance of medication adherence.

## Quality of included studies

All RCTs in this study were at high risk of bias due to inadequate blinding, as the nature of the intervention—training doctors in communication skills—makes effective blinding challenging. Inadequate blinding can introduce subjectivity into assessments, potentially exaggerating the intervention's effects or leading to false positives, which limits the generalizability of the conclusions. Furthermore, only two studies provided information on allocation concealment, highlighting another potential source of bias. Allocation concealment is essential for preventing selection bias by ensuring that participant assignment to treatment groups is unpredictable and unrelated to participant characteristics. A lack of transparency in this process can lead to biased participant selection, further undermining the validity of the research findings. Future researchers should address these issues more thoroughly in the design and reporting stages to improve overall research quality. Additionally, several cross-sectional studies failed to meet key quality assessment criteria, such as evaluator blinding, quality assurance measures, patient exclusion explanations, handling of missing data, and completeness of follow-up data. This can compromise the objectivity, data consistency, sample representativeness, analysis accuracy, and long-term effectiveness evaluation of research outcomes. However, the quality of the cohort studies included in this review was generally high. Despite our efforts to mitigate the impact of these biases through rigorous screening and evaluation processes, variations in bias risks among different studies may still lead to inconsistent conclusions.

## Limitations

Our study had several limitations. First, the quality of the included studies was variable, although most were rated as adequate. Second, most of the included observational studies were cross-sectional, limiting the ability to infer causality. In this systematic review, the majority of the observational studies included are cross-sectional. A primary limitation of this study design is its inability to establish temporal precedence, which in turn limits our ability to infer causal relationships. Since cross-sectional studies collect data at a single point in time, we cannot determine whether the observed associations are causal. Third, the diverse study settings and populations limit the generalizability of the findings. The studies included in this systematic review were primarily conducted in the United States, which may limit the global applicability of our findings. Particularly considering the potential cultural differences in doctor-patient communication, the research environment in the United States may not fully represent the medical practices and patient experiences worldwide. Fourth, although we minimized the exclusion of literature due to the unavailability of full texts, any remaining unobtainable documents, if they contained significant data or perspectives crucial to our research question, might impact the comprehensiveness and accuracy of our conclusions. Given these limitations, while our

findings provide valuable insights, they need to be validated within a broader and more rigorous research context.

## CONCLUSION

In conclusion, this review underscores the importance of strong doctor–patient communication in fostering better medication adherence and blood pressure control. However, further research may be needed to address variations and nuances in the literature and to establish clear implications for clinical practice. Moreover, in clinical practice, strategies to enhance doctor–patient communication should be incorporated, given the potential to improve medication adherence and blood pressure control among hypertensive patients.

## ACKNOWLEDGEMENTS

We thank all of the authors of the included studies.

### Funding

The authors received no funding for this work.

### Competing Interests

The authors declare that they have no competing interests.

### Author Contributions

- Jianwei Zeng conceived and designed the experiments, performed the experiments, analyzed the data, prepared figures and/or tables, authored or reviewed drafts of the article, and approved the final draft.
- Yuqiang Gao analyzed the data, prepared figures and/or tables, and approved the final draft.
- Chen Hou conceived and designed the experiments, performed the experiments, authored or reviewed drafts of the article, and approved the final draft.
- Tao Liu analyzed the data, authored or reviewed drafts of the article, and approved the final draft.

### Data Availability

This is a systematic review.

### Supplemental Information

Supplemental information for this article can be found online at http://dx.doi.org/10.7717/peerj.18527#supplemental-information.

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
