# Peer review of "The impact of doctor–patient communication on medication adherence and blood pressure control in patients with hypertension: a systematic review"

_PeerJ, doi:10.7717/peerj.18527_

## Round 0.1 · original submission · Major Revisions

Dear authors,

We kindly request that you carefully review the comments provided by the reviewers. Their valuable suggestions offer insights to enhance your manuscript. Incorporate their suggestions and carefully address all comments in your manuscript; it will significantly strengthen its content.

Reviewer 1 ·

Basic reporting

The abstract could better summarize key findings and explicitly state the systematic review's implications for practice and policy.

Some parts of the discussion could be condensed to improve clarity and conciseness, ensuring the most critical information stands out more effectively.

Experimental design

While the eligibility criteria are outlined, the manuscript could benefit from a more detailed description of the study selection process to ensure reproducibility.

More explicit details on the conflict resolution process during data extraction could enhance the transparency and reliability of the results.

Validity of the findings

The discussion of the limitations is somewhat brief. Expanding on the potential biases and the heterogeneity in study designs would provide a more balanced view of the findings.

The implications for practice are mentioned, but a more detailed discussion on how these findings can be implemented in clinical settings would be beneficial.

Additional comments

The revisions should aim to enhance the clarity of the abstract, expand the discussion on limitations, and provide more detailed implications for clinical practice as mentioned in the review. Once these areas are addressed, the manuscript will be a valuable contribution to the literature on doctor-patient communication and its impact on hypertension management.

Reviewer 2 ·

Basic reporting

The manuscript presents a comprehensive systematic review on the impact of doctor-patient communication on medication adherence and blood pressure control in patients with hypertension.

Experimental design

The manuscript would benefit from a more detailed discussion on the quality of the included studies. While the use of tools like the Newcastle-Ottawa Scale and the Cochrane bias tool is mentioned, specific issues or biases detected in these studies should be discussed to inform the reader of potential limitations affecting the review’s outcomes.
The narrative synthesis approach is appropriate given the heterogeneity of studies. However, the manuscript should better address how this heterogeneity impacts the overall conclusions that can be drawn. It would be advantageous to see a thematic analysis of how different aspects of communication (e.g., style, content) influence outcomes across various studies.

Validity of the findings

No comments.

Additional comments

No comments.

Reviewer 3 ·

Basic reporting

The background provided is comprehensive and demonstrates a clear understanding of the research area. The manuscript follows a conventional structure for a systematic review, which facilitates the comprehension of the study's aims, methods, results, and conclusions. However, several areas in the formatting and detail of reporting need further attention to meet the publication standards. Below are some detailed comments:

Line 42, Line 76, Line 80,… You did not number the Instruction section but numbered the Methods section and later sections. Please be consistent.

Line 45-47 Please list the statistics on how the global prevalence of hypertension changed. What is the rate in 1990 and what is the rate in 2019? What is the rate now? Please cite more recent papers as well if possible.

Line 49-51
1. Do you mean 47% of women and 38% of men were under treatment, instead of being treated?
2. Please report the statistics with one digit to match the precision of other reported statistics.

Line 60-62 Please provide a citation to this sentence. Please clarify how medication adherence is measured and what is considered poor medication adherence.

Line 86-105: To enhance clarity and readability, consider reporting your search strategies using a summary table. This approach will not only streamline the presentation of your methodologies but also facilitate the reproducibility of your research.

Line 144: Please be consistent with the wording of the study excluded reason as used in Figure 1.

Line 166 You should provide more details of scales and questionnaires being used in the results section, rather than the discussion section in Line 233-244. That information should belong to the result.

Line 174: Please clarify what is primary healthcare information intervention. You should add “to patients” at the end of this sentence.

Line 182: there is an extra “.” In “Naike et al26.”. A similar issue is in Line 195 and Line 198.

Line 184, Line 185: Please report the effects using appropriate effect measures. The interpretation of beta coefficients varies depending on the statistical methods employed in their respective studies. For instance, the interpretation of beta coefficients differs significantly between logistic regression and linear regression. It is crucial to specify the type of analysis used to ensure a clear and accurate interpretation of the results.

Line 189: Please provide more details regarding the informative intervention and how this enhances blood pressure control.

Line 191: Please clarify how patient-centered engagement is measured. Providing detailed information on the methods used to assess this engagement will greatly benefit the reader and enhance the transparency of your findings.

Line 206: there is an extra “.”

Line 207: Please clarify patient communication and medication adherence are measured. Providing detailed information on the methods used to assess this engagement will greatly benefit the reader and enhance the transparency of your findings.

Line 230-234 Please add a discussion of how the two points impact the conclusion and validity of your study.

Line 241-244 How long the studies are followed should be in the results section instead of the discussion section.

Line 268: it should be “of included studies”

Line 367 shows a citation error

Line 291-293 Please clarify where Tinsel, Cho, and Mahmoudian’s study are also only in black? Please add the discussion in the general population as well

Line 300: Please clarify and add more detailed information to the survey, such as where and when the survey happened. What type of survey was this study? What kind of population was included in this survey?

Table 4 There is an extra “,” in "Total" column.

Experimental design

The original primary research is within the aim of the scope of the journal. The research question is well-defined, relevant, and meaningful. The methods described are with sufficient detail and information to replicate. Here are some detailed comments:

Line 86-105: Was the same search strategy applied across all five databases? It is crucial to confirm whether each database supports the same search logic to ensure that your strategy is uniformly effective. If the search logic varied among the databases, please detail the specific search strategies used for each one to facilitate the reproducibility of your research.

Line 120-121: You mentioned that in cases of disagreement, a third researcher (TL) was consulted. Could you please clarify how the final decision is determined? If TL is responsible for making the final decision, it would be beneficial to explicitly state this to ensure the clarity of the decision-making process.

Validity of the findings

The methods employed are suitable for the study design, and the results have been reported clearly. However, there are several issues that may challenge the validity of the findings and need to be addressed. The most significant concerns include the exclusion of 33% of studies due to the unavailability of full texts and the lack of clarity in the search strategy. These issues should be thoroughly resolved to strengthen the credibility of the study's conclusions. Here are some detailed comments:

Line 84: You noted that your database search extended up to January 27, 2024. However, the results section only includes studies conducted up to 2020. Could you please confirm whether there were no relevant studies published on this topic after 2020? If there are newer studies, they should be incorporated to ensure the review’s comprehensiveness and currency.

Line 88-105: There appear to be errors and omissions in your search strategy that could significantly impact your results. For instance, in line 96, the term “Personnel, Communications”[Title/Abstract] is missing a closing parenthesis. This oversight could dramatically alter the search results. Similar issues with missing parentheses are also evident in lines 97 and 105. Please review and correct these discrepancies to ensure the accuracy and reliability of your search strategy.

Line 158: You mentioned that two RCTs were deemed to have a high risk of bias. Please discuss the potential impact this might have on the validity of the study's conclusions. It is crucial to understand how these biases could affect the overall findings and whether they might skew the interpretation of the data.

Line 215-219, Line 227 You listed three studies that reported null results regarding the relationship between communication and medical adherence. And you mentioned that in general, the results suggest a promising link. Please discuss why those are reported in negative results and how those negative results will impact the validity of your conclusion that the results suggested a promising link. Please discuss this in the discussion section.

Line 245-246 You mentioned that the training period was the factor that why the intervention failed. However, no information about how long the training period of each study is mentioned in this study. Please support this conclusion using statistics.

Figure 1: It is noted that 33% (19 out of 57) of publications were excluded because the full texts were not found. This represents a significant portion of the publications, which could introduce a selection bias to your conclusions. Please explain the reasons why these full texts were unavailable and discuss any potential impacts this might have on the results of your study.

---

## Round 0.2 · Major Revisions

Dear authors,

We kindly request that you carefully review the comments provided by the reviewers. Their valuable suggestions offer insights to enhance your manuscript. Incorporate their suggestions and carefully address all comments in your manuscript; it will significantly strengthen its content. Thanks

Reviewer 1 ·

Basic reporting

The author made changes as suggested, hence the manuscript could be accepted now.

Experimental design

The author made changes as suggested, hence the manuscript could be accepted now.

Validity of the findings

The author made changes as suggested, hence the manuscript could be accepted now.

Reviewer 2 ·

Basic reporting

The authors included 18 observational studies and 7 RCTs, RCTs were rated as having a high risk of bias due to lack of blinding, and only 2 reported allocation concealment. For observational studies, 6 were low risk and 7 moderate risk of bias. The authors should discuss how the variable quality of included studies impacts the strength and reliability of their conclusions.

Experimental design

There is substantial heterogeneity in how communication and medication adherence were measured across studies. A more detailed discussion of the specific measures used and their comparability would strengthen the review.

Most included observational studies were cross-sectional. The authors should more explicitly discuss this limitation and caution against over-interpreting associations as causal relationships.

Validity of the findings

Some studies found significant associations between communication and outcomes while others did not. The authors should provide a more nuanced discussion of these inconsistencies, exploring potential reasons for differing results (e.g. study design, population characteristics, specific communication interventions).

The included studies were conducted primarily in the USA, The authors should comment on how this affects the global applicability of their findings, especially given potential cultural differences in doctor-patient communication.

Reviewer 3 ·

Basic reporting

Thank you for inviting me to review this manuscript again. The authors have successfully addressed my previous comments regarding basic reporting. I have no further questions.

Experimental design

Thank you for inviting me to review this manuscript again. The authors have successfully addressed my previous comments regarding experimental design. I have no further questions.

Validity of the findings

In my previous review, in the first draft, you mentioned in Figure 1 that after screening titles and abstracts for eligibility(n=550), 57 publications left and were examined full text, however, 19 out of 57 publications were excluded because the full texts were not found. I commented my concerns that 33% (19 out of 57) of publication represents a significant portion of the publications, which could introduce a selection bias to your conclusions.

In this revision, you have please response that you successfully obtained the full texts of the vast majority of publications that were originally unavailable for various reasons. And only one publication (an English thesis) whose full text cannot be directly accessed exists.

However, in Figure 1 of your revised manuscript, it now states that after screening titles and abstracts for eligibility, 41 publications were left and examined in full text. Please clarify why there is a discrepancy between the initial 57 eligible publications after reviewing titles and abstracts and the current 41 eligible publications after the same review process.

---

## Round 0.3 · accepted · Accept

Dear Dr. Liu,

Thank you for submitting the revised version of your manuscript. After a thorough review of the changes by the reviewers and myself, I am pleased to inform you that all the reviewers' comments have been adequately addressed. Therefore, your manuscript is ready for publication in PeerJ.

I thank all reviewers for their efforts in improving the manuscript and the authors' cooperation throughout the review process.

Sincerely yours,
Stefano Menini

Reviewer 2 ·

Basic reporting

The authors have resolved my concerns.

Experimental design

The authors have resolved my concerns.

Validity of the findings

The authors have resolved my concerns.

Additional comments

The authors have resolved my concerns.